# NEURAL TANGENTS:
## FAST AND EASY INFINITE NEURAL NETWORKS IN PYTHON

**Roman Novak**[*]  **Lechao Xiao**[*]  **Jiri Hron**[†],  **Jaehoon Lee,**

**Alexander A. Alemi,**  **Jascha Sohl-Dickstein,**  **Samuel S. Schoenholz**[*]

Google Brain,  [†]University of Cambridge

{romann, xlc}@google.com, jh2084@cam.ac.uk, {jaehlee, alemi, jaschasd, schsam}@google.com

## ABSTRACT

NEURAL TANGENTS is a library for working with infinite-width neural networks. It provides a high-level API for specifying complex and hierarchical neural network architectures. These networks can then be trained and evaluated either at finite-width as usual or in their infinite-width limit. Infinite-width networks can be trained analytically using exact Bayesian inference or using gradient descent via the Neural Tangent Kernel. Additionally, NEURAL TANGENTS provides tools to study gradient descent training dynamics of wide but finite networks in either function space or weight space.

The entire library runs out-of-the-box on CPU, GPU, or TPU. All computations can be automatically distributed over multiple accelerators with near-linear scaling in the number of devices. NEURAL TANGENTS is available at

www.github.com/google/neural-tangents

We also provide an accompanying interactive **Colab notebook**[1].

## 1 INTRODUCTION

Deep neural networks (DNNs) owe their success in part to the broad availability of high-level, flexible, and efficient software libraries like Tensorflow (Abadi et al., 2015), Keras (Chollet et al., 2015), PyTorch.nn (Paszke et al., 2017), Chainer (Tokui et al., 2015; Akiba et al., 2017), JAX (Bradbury et al., 2018a), and others. These libraries enable researchers to rapidly build complex models by constructing them out of smaller primitives. The success of new machine learning approaches will similarly depend on developing sophisticated software tools to support them.

### 1.1 INFINITE-WIDTH BAYESIAN NEURAL NETWORKS

Recently, a new class of machine learning models has attracted significant attention, namely, deep *infinitely wide* neural networks. In the infinite-width limit, a large class of Bayesian neural networks become Gaussian Processes (GPs) with a specific, architecture-dependent, compositional kernel; these models are called Neural Network Gaussian Processes (NNGPs). This correspondence was first established for shallow fully-connected networks by Neal (1994) and was extended to multi-layer setting in (Lee et al., 2018; Matthews et al., 2018b). Since then, this correspondence has been expanded to a wide range of nonlinearities (Matthews et al., 2018a; Novak et al., 2019) and architectures including those with convolutional layers (Garriga-Alonso et al., 2019; Novak et al., 2019), residual connections (Garriga-Alonso et al., 2019), pooling (Novak et al., 2019), as well as graph neural networks (Du et al., 2019). The results for individual architectures have subsequently been generalized, and it was shown that a GP correspondence holds for a general class of networks that can be mapped to so-called *tensor programs* in (Yang, 2019). The recurrence relationship defining

---

[*]Equal contribution. [†]Work done during an internship at Google Brain.

[1]colab.research.google.com/github/google/neural-tangents/blob/master/notebooks/neural_tangents_cookbook.ipynb

the NNGP kernel has additionally been extensively studied in the context of mean field theory and initialization (Cho & Saul, 2009; Daniely et al., 2016; Poole et al., 2016; Schoenholz et al., 2016; Yang & Schoenholz, 2017; Xiao et al., 2018; Li & Nguyen, 2019; Pretorius et al., 2018; Hayou et al., 2018; Karakida et al., 2018; Blumenfeld et al., 2019; Hayou et al., 2019).

## 1.2 INFINITE-WIDTH NEURAL NETWORKS TRAINED BY GRADIENT DESCENT

In addition to enabling a closed form description of Bayesian neural networks, the infinite-width limit has also very recently provided insights into neural networks trained by gradient descent. In the last year, several papers have shown that randomly initialized neural networks trained with gradient descent are characterized by a distribution that is related to the NNGP, and is described by the so-called Neural Tangent Kernel (NTK) (Jacot et al., 2018; Lee et al., 2019; Chizat et al., 2019), a kernel which was implicit in some earlier papers (Li & Liang, 2018; Allen-Zhu et al., 2018; Du et al., 2018a;b; Zou et al., 2019). In addition to this "function space" perspective, a dual, "weight space" view on the wide network limit was proposed in Lee et al. (2019) which showed that networks under gradient descent were well-described by the first-order Taylor series about their initial parameters.

## 1.3 PROMISE AND PRACTICAL BARRIERS TO WORKING WITH INFINITE-WIDTH NETWORKS

Combined, these discoveries established infinite-width networks as useful theoretical tools to understand a wide range of phenomena in deep learning. Furthermore, the practical utility of these models has been proven by achieving state-of-the-art performance on image classification benchmarks among GPs without trainable kernels (Garriga-Alonso et al., 2019; Novak et al., 2019; Arora et al., 2019a), and by their ability to match or exceed the performance of finite width networks in some situations, especially for fully- and locally-connected model families (Lee et al., 2018; Novak et al., 2019; Arora et al., 2019b).

However, despite their utility, using NNGPs and NTK-GPs is arduous and can require weeks-to-months of work by seasoned practitioners. Kernels corresponding to neural networks must be derived by hand on a per-architecture basis. Overall, this process is laborious and error prone, and is reminiscent of the state of neural networks before high quality Automatic Differentiation (AD) packages proliferated.

## 1.4 SUMMARY OF CONTRIBUTIONS

In this paper, we introduce a new open-source software library called NEURAL TANGENTS targeting JAX (Bradbury et al., 2018a) to accelerate research on infinite limits of neural networks. The main features of NEURAL TANGENTS are:[2]

- A high-level neural network API for specifying complex, hierarchical, models. Networks specified using this API can have their infinite-width NNGP kernel and NTK evaluated analytically (§2.1, Listings 1, 2, 3, §B.2).
- Functions to approximate infinite-width kernels by Monte Carlo sampling for networks whose kernels cannot be constructed analytically. These methods are agnostic to the neural network library used to build the network and are therefore quite versatile (§2.2, Figure 2, §B.5).
- An API to analytically perform inference using infinite-width networks either by computing the Bayesian posterior or by computing the result of continuous gradient descent with an MSE loss. The API additionally includes tools to perform inference by numerically solving the ODEs corresponding to: continuous gradient descent, with-or-without momentum, on arbitrary loss functions, at finite or infinite time (§2.1, Figure 1, §B.4).
- Functions to compute arbitrary-order Taylor series approximations to neural networks about a given setting of parameters to explore the weight space perspective on the infinite-width limit (§B.6, Figure 6).
- Leveraging XLA, our library runs out-of-the-box on CPU, GPU, or TPU. Kernel computations can automatically be distributed over multiple accelerators with near-perfect scaling (§3.2, Figure 5, §B.3).

---

[2]See §A for NEURAL TANGENTS comparison against specific relevant prior works.

We begin with three short examples (§2) that demonstrate the ease, efficiency, and versatility of performing calculations with infinite networks using NEURAL TANGENTS. With a high level view of the library in hand, we then dive into a number of technical aspects of our library (§3).

## 1.5 BACKGROUND

We briefly describe the NNGP (§1.1) and NTK (§1.2). **NNGP.** Neural networks are often structured as affine transformations followed by pointwise applications of nonlinearities. Let $z_i^l(x)$ describe the $i^{\text{th}}$ pre-activation following a linear transformation in $l^{\text{th}}$ layer of a neural network. At initialization, the parameters of the network are randomly distributed and so central-limit theorem style arguments can be used to show that the pre-activations become Gaussian distributed with mean zero and are therefore described entirely by their covariance matrix $\mathcal{K}(x, x') = \mathbb{E}[z_i^l(x) z_i^l(x')]$. This describes a NNGP with the kernel, $\mathcal{K}(x, x')$. One can use the NNGP to make Bayesian posterior predictions at a test point, $x$, which are Gaussian distributed with with mean $\mu(x) = \mathcal{K}(x, \mathcal{X})\mathcal{K}(\mathcal{X}, \mathcal{X})^{-1}\mathcal{Y}$ and variance $\sigma^2(x) = \mathcal{K}(x, x) - \mathcal{K}(x, \mathcal{X})\mathcal{K}(\mathcal{X}, \mathcal{X})^{-1}\mathcal{K}(\mathcal{X}, x)$, where $(\mathcal{X}, \mathcal{Y})$ is the training set of inputs and targets respectively. **NTK.** When neural networks are optimized using continuous gradient descent with learning rate $\eta$ on mean squared error (MSE) loss, the function evaluated on training points evolves as $\partial_t f_t(\mathcal{X}) = -\eta J_t(\mathcal{X}) J_t(\mathcal{X})^T (f_t(\mathcal{X}) - \mathcal{Y})$ where $J_t(\mathcal{X})$ is the Jacobian of the output $f_t$ evaluated at $\mathcal{X}$ and $\Theta_t(\mathcal{X}, \mathcal{X}) = J_t(\mathcal{X}) J_t(\mathcal{X})^T$ is the NTK. In the infinite-width limit, the NTK remains constant ($\Theta_t = \Theta$) throughout training and the time-evolution of the outputs can be solved in closed form as a Gaussian with mean $f_t(x) = \Theta(x, \mathcal{X})\Theta(\mathcal{X}, \mathcal{X})^{-1} (I - \exp\left[-\eta\Theta(\mathcal{X}, \mathcal{X})t\right]) \mathcal{Y}$.

## 2 EXAMPLES

We begin by applying NEURAL TANGENTS to several example tasks. While these tasks are designed for pedagogy rather than research novelty, they are nonetheless emblematic of problems regularly faced in research. We emphasize that without NEURAL TANGENTS, it would be necessary to derive the kernels for each architecture *by hand*.

### 2.1 INFERENCE WITH AN INFINITELY WIDE NEURAL NETWORK

We begin by training an infinitely wide neural network with gradient descent and comparing the result to training an ensemble of wide-but-finite networks. This example is worked through in detail in the **Colab notebook**.[3]

We train on a synthetic dataset with training data drawn from the process $y_i = \sin(x_i) + \epsilon_i$ with $x_i \sim \text{Uniform}(-\pi, \pi)$ and $\epsilon_i \sim \mathcal{N}(0, \sigma^2)$ independently and identically distributed. To train an infinite neural network with $\text{Erf}$ activations[4] on this data using gradient descent and an MSE loss we write the following:

```python
from neural_tangents import predict, stax

init_fn, apply_fn, kernel_fn = stax.serial(
    stax.Dense(2048, W_std=1.5, b_std=0.05), stax.Erf(),
    stax.Dense(2048, W_std=1.5, b_std=0.05), stax.Erf(),
    stax.Dense(1, W_std=1.5, b_std=0.05))

y_mean, y_var = predict.gp_inference(kernel_fn, x_train, y_train, x_test, 'ntk',
                                     diag_reg=1e-4, compute_cov=True)
```

The above code analytically generates the predictions that would result from performing gradient descent for an infinite amount of time. However, it is often desirable to investigate finite-time learning dynamics of deep networks. This is also supported in NEURAL TANGENTS as illustrated in the following snippet:

---

[3]www.colab.research.google.com/github/google/neural-tangents/blob/master/notebooks/neural_tangents_cookbook.ipynb

[4]Error function, a nonlinearity similar to $\tanh$; see §D for other implemented nonlinearities, including $\text{Relu}$.

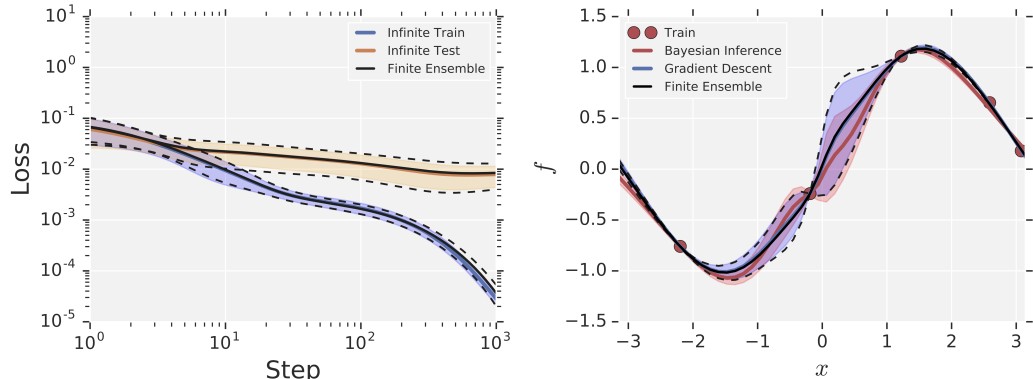

Figure 1: **Training dynamics for an ensemble of finite-width networks compared with an infinite network. Left:** Mean and variance of the train and test MSE loss evolution throughout training. **Right:** Comparison between the predictions of the trained infinite network and the respective ensemble of finite-width networks. The shaded region and the dashed lines denote two standard deviations of uncertainty in the predictions for the infinite network and the ensemble respectively.

```
predict_fn = predict.gradient_descent_mse_gp(kernel_fn, x_train, y_train, x_test, 'ntk',
                                        diag_reg=1e-4, compute_cov=True)
y_mean, y_var = predict_fn(t=100)  # Predict the distribution at t = 100.
```

The above specification set the hidden layer widths to 2048, which has no effect on the infinite width network inference, but the `init_fn` and `apply_fn` here correspond to ordinary finite width networks. In Figure 1 we compare the result of this exact inference with training an ensemble of one-hundred of these finite-width networks by looking at the training curves and output predictions of both models. We see excellent agreement between exact inference using the infinite-width model and the result of training an ensemble using gradient descent.

## 2.2 AN INFINITELY WIDERESNET

The above example considers a relatively simple network on a synthetic task. In practice we may want to consider real-world architectures, and see how close they are to their infinite-width limit. For this task we study a variant of an infinite-channel Wide Residual Network (Zagoruyko & Komodakis, 2016) (WRN-28-∞). We first define both finite and infinite models within Listing 1.

We now study how quickly the kernel of the finite-channel WideResNet approaches its infinite channel limit. We explore two different axes along which convergence takes place: first, as a function of the number of channels (as measured by the widening factor, $k$) and second as a function of the number of finite-network Monte Carlo samples we average over. NEURAL TANGENTS makes it easy to compute MC averages of finite kernels using the following snippet:

```
kernel_fn = nt.monte_carlo_kernel_fn(init_fn, apply_fn, rng_key, n_samples)
sampled_kernel = kernel_fn(x, x)
```

The convergence is shown in Figure 2. We see that as both the number of samples is increased or the network is made wider, the empirical kernel approaches the kernel of the infinite network. As noted in Novak et al. (2019), for any finite widening factor the MC estimate is biased. Here, however, the bias is small relative to the variance and the distance to the empirical kernel decreases with the number of samples.

## 2.3 COMPARISON OF NEURAL NETWORK ARCHITECTURES AND TRAINING SET SIZES

The above examples demonstrate how one might construct a complicated architecture and perform inference using NEURAL TANGENTS Next we train a range of architectures on CIFAR-10 and compare their performance as a function of dataset size. In particular, we compare a fully-connected network,

a convolutional network whose penultimate layer vectorizes the image, and the wide-residual network described above. In each case, we perform exact infinite-time inference using the analytic infinite-width NNGP or NTK. For each architecture we perform a hyperparameter search over the depth of the network, selecting the depth that maximizes the marginal log likelihood on the training set.

```python
def WideResNetBlock(channels, strides=(1, 1), channel_mismatch=False):
  Main = stax.serial(stax.Relu(), stax.Conv(channels, (3, 3), strides, padding='SAME'),
                     stax.Relu(), stax.Conv(channels, (3, 3), padding='SAME'))
  Shortcut = (stax.Identity() if not channel_mismatch else
             stax.Conv(channels, (3, 3), strides, padding='SAME'))
  return stax.serial(stax.FanOut(2), stax.parallel(Main, Shortcut), stax.FanInSum())

def WideResNetGroup(n, channels, strides=(1, 1)):
  blocks = [WideResNetBlock(channels, strides, channel_mismatch=True)]
  for _ in range(n - 1):
    blocks += [WideResNetBlock(channels, (1, 1))]
  return stax.serial(*blocks)

def WideResNet(block_size, k, num_classes):
  return stax.serial(stax.Conv(16, (3, 3), padding='SAME'),
                     WideResNetGroup(block_size, int(16 * k)),
                     WideResNetGroup(block_size, int(32 * k), (2, 2)),
                     WideResNetGroup(block_size, int(64 * k), (2, 2)),
                     stax.GlobalAvgPool(), stax.Dense(num_classes))

init_fn, apply_fn, kernel_fn = WideResNet(block_size=4, k=1, num_classes=10)
```

Listing 1: **Definition of an infinitely WideResNet**. This snippet simultaneously defines a finite ( `init_fn`, `apply_fn` ) and an infinite ( `kernel_fn` ) model. This model is used in Figures 2 and 3.

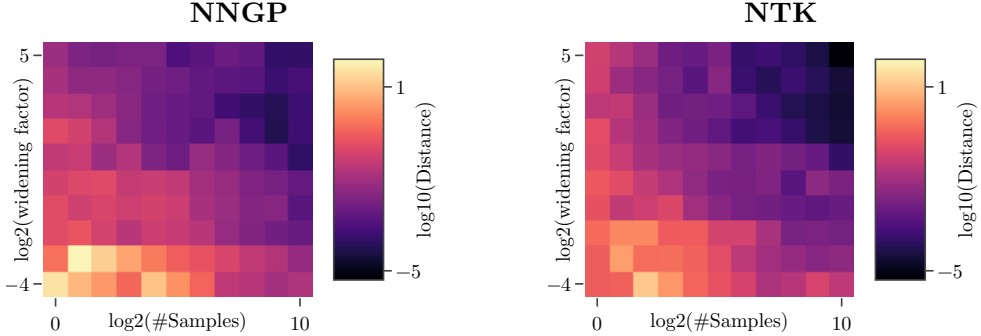

Figure 2: **Convergence of the Monte Carlo (MC) estimates** of the WideResNet WRN-28-$k$ (where $k$ is the widening factor) NNGP and NTK kernels (computed with `monte_carlo_kernel_fn` ) to their analytic values (WRN-28-$\infty$, computed with `kernel_fn` ), as the network gets wider by increasing the widening factor (vertical axis) and as more random networks are averaged over (horizontal axis). **Experimental detail.** The kernel is computed in 32-bit precision on a $100 \times 50$ batch of $8 \times 8$-downsampled CIFAR10 (Krizhevsky, 2009) images. For sampling efficiency, for NNGP the output of the penultimate layer was used, and for NTK the output layer was assumed to be of dimension 1 (all logits are i.i.d. conditioned on a given input). The displayed distance is the relative Frobenius norm squared, i.e. $\|\mathcal{K} - \mathcal{K}_{k,n}\|_{\mathrm{F}}^2 / \|\mathcal{K}\|_{\mathrm{F}}^2$, where $k$ is the widening factor and $n$ is the number of samples.

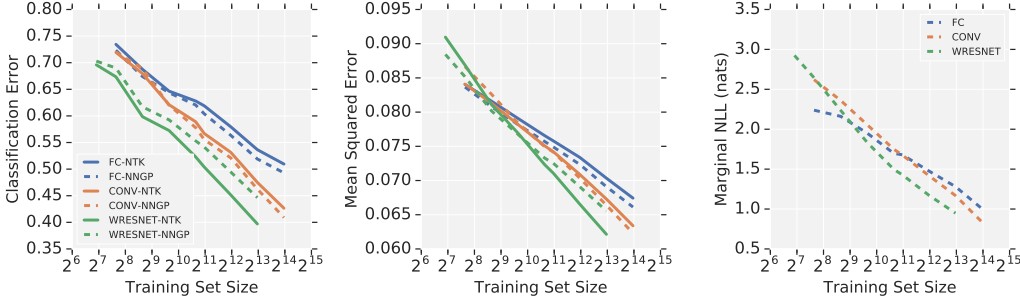

Figure 3: **CIFAR-10 classification with varying neural network architectures.** NEURAL TAN-GENTS simplify experimentation with architectures. Here we use infinite time NTK inference and full Bayesian NNGP inference for CIFAR-10 for **Fully Connected (FC, Listing 3)**, **Convolutional network without pooling (CONV, Listing 2)**, and **Wide Residual Network w/ pooling (WRES-NET, Listing 1)**. As is common in prior work (Lee et al., 2018; Novak et al., 2019), the classification task is treated as MSE regression on zero-mean targets like $(-0.1, \ldots, -0.1, 0.9, -0.1, \ldots, -0.1)$. For each training set size, the best model in the family is selected by minimizing the mean negative marginal log-likelihood (NLL, right) on the training set.

The results are shown in Figure 3. We see that in each case the performance of the model increases approximately logarithmically in the size of the dataset. Moreover, we observe a clear hierarchy of performance, especially at large dataset size, in terms of architecture (FC < CONV < WRESNET w/ pooling).

## 3 IMPLEMENTATION: TRANSFORMING TENSOR OPS TO KERNEL OPS

Neural networks are compositions of basic tensor operations such as: dense or convolutional affine transformations, application of pointwise nonlinearities, pooling, or normalization. For most networks without weight tying between layers the kernel computation can also be written compositionally and there is a direct correspondence between tensor operations and kernel operations (see §3.1 for an example). The core logic of NEURAL TANGENTS is a set of *translation rules*, that sends each tensor operation acting on a finite-width layer to a corresponding transformation of the kernel for an infinite-width network. This is illustrated in Figure 4 for a simple convolutional architecture. In the associated table, we compare tensor operations (second column) with corresponding transformations of the NT and NNGP kernel tensors (third and fourth column respectively). See §D for a list of all tensor operations for which translation rules are currently implemented.

One subtlety to consider when designing networks is that most infinite-width results require nonlinear transformations to be preceded by affine transformations (either dense or convolutional). This is because infinite-width results often assume that the pre-activations of nonlinear layers are approximately Gaussian. Randomness in weights and biases causes the output of infinite affine layers to satisfy this Gaussian requirement. Fortunately, prefacing nonlinear operations with affine transformations is common practice when designing neural networks and NEURAL TANGENTS will raise an error if this requirement is not satisfied.

### 3.1 A TASTE OF TENSOR-TO-KERNEL OPS TRANSLATION

To get some intuition behind the translation rules, we consider the case of a nonlinearity followed by a dense layer. Let $z = z(\mathcal{X}, \theta) \in \mathbb{R}^{d \times n}$ be the preactivations resulting from $d$ distinct inputs at a node in some hidden layer of a neural network. Suppose $z$ has NNGP kernel and NTK given by

$$\mathcal{K}_z = \mathbb{E}_\theta \left[ z_i z_i^T \right], \quad \Theta_z = \mathbb{E}_\theta \left[ \frac{\partial z_i}{\partial \theta} \left( \frac{\partial z_i}{\partial \theta} \right)^T \right] \tag{1}$$

where $z_i \in \mathbb{R}^d$ is the $i^{\text{th}}$ neuron and $\theta$ are the parameters in the network up until $z$. Here $d$ is the cardinality of the network inputs $\mathcal{X}$ and $n$ is the number of neurons in the $z$ node. We

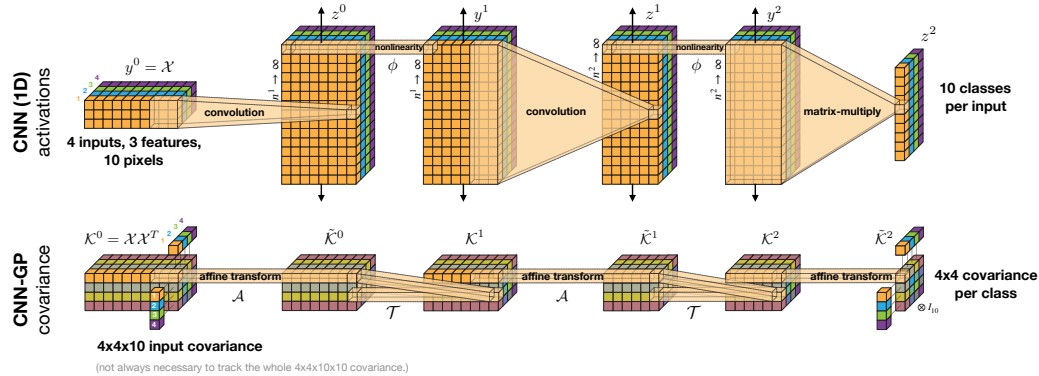

| Layer | Tensor Op | NNGP Op | NTK Op |
|---|---|---|---|
| **0** (input) | $y^0 = \mathcal{X}$ | $\mathcal{K}^0 = \mathcal{X}\mathcal{X}^T$ | $\Theta^0 = 0$ |
| **0** (pre-activations) | $z^0 = \text{Conv}\left(y^0\right)$ | $\tilde{\mathcal{K}}^0 = \mathcal{A}\left(\mathcal{K}^0\right)$ | $\tilde{\Theta}^0 = \tilde{\mathcal{K}}^0 + \mathcal{A}\left(\Theta^0\right)$ |
| **1** (activations) | $y^1 = \phi\left(z^0\right)$ | $\mathcal{K}^1 = \mathcal{T}\left(\tilde{\mathcal{K}}^0\right)$ | $\Theta^1 = \dot{\mathcal{T}}\left(\tilde{\mathcal{K}}^0\right) \odot \tilde{\Theta}^0$ |
| **1** (pre-activations) | $z^1 = \text{Conv}\left(y^1\right)$ | $\tilde{\mathcal{K}}^1 = \mathcal{A}\left(\mathcal{K}^1\right)$ | $\tilde{\Theta}^1 = \tilde{\mathcal{K}}^1 + \mathcal{A}\left(\Theta^1\right)$ |
| **2** (activations) | $y^2 = \phi\left(z^1\right)$ | $\mathcal{K}^2 = \mathcal{T}\left(\tilde{\mathcal{K}}^1\right)$ | $\Theta^2 = \dot{\mathcal{T}}\left(\tilde{\mathcal{K}}^1\right) \odot \tilde{\Theta}^1$ |
| **2** (readout) | $z^2 = \text{Dense} \circ \text{Flatten}\left(y^2\right)$ | $\tilde{\mathcal{K}}^2 = \text{Tr}\left(\mathcal{K}^2\right)$ | $\tilde{\Theta}^2 = \tilde{\mathcal{K}}^2 + \text{Tr}\left(\Theta^2\right)$ |

Figure 4: **An example of the translation of a convolutional neural network into a sequence of kernel operations.** We demonstrate how the compositional nature of a typical NN computation on its inputs induces a corresponding compositional computation on the NNGP and NT kernels. Presented is a 2-hidden-layer 1D CNN with nonlinearity $\phi$, performing regression on the 10-dimensional outputs $z^2$ for each of the 4 (**1**, **2**, **3**, **4**) inputs $x$ from the dataset $\mathcal{X}$. To declutter notation, unit weight and zero bias variances are assumed in all layers. **Top:** recursive output ($z^2$) computation in the CNN (top) induces a respective recursive NNGP kernel ($\tilde{\mathcal{K}}^2 \otimes I_{10}$) computation (NTK computation being similar, not shown). **Bottom:** explicit listing of tensor and corresponding kernel ops in each layer. See Table 1 for operation definitions. Illustration and description adapted from Figure 3 in Novak et al. (2019).

assume $z$ is a mean zero multivariate Gaussian. We wish to compute the kernel corresponding to $h = \text{Dense}\left(\sigma_\omega, \sigma_b\right)\left(\phi(z)\right)$ by computing the kernels of $y = \phi(z)$ and $h = \text{Dense}\left(\sigma_\omega, \sigma_b\right)(y)$ separately. Here,

$$h = \text{Dense}(\sigma_\omega, \sigma_b)(y) \equiv \left(1/\sqrt{n}\right)\sigma_\omega W y + \sigma_b \beta, \tag{2}$$

and the variables $W_{ij}$ and $\beta_i$ are i.i.d. Gaussian $\mathcal{N}(0,1)$. We will compute kernel operations - denoted $\phi^*$ and $\text{Dense}(\sigma_\omega, \sigma_b)^*$ - induced by the tensor operations $\phi$ and $\text{Dense}(\sigma_\omega, \sigma_b)$ [5]. Finally, we will compute the kernel operation associated with the composition $\left(\text{Dense}(\sigma_\omega, \sigma_b) \circ \phi\right)^* = \text{Dense}(\sigma_\omega, \sigma_b)^* \circ \phi^*$.

First we compute the NNGP and NT kernels for $y$. To compute $\mathcal{K}_y$ note that from its definition,

$$\mathcal{K}_y = \mathcal{K}_{\phi(z)} = \mathbb{E}_\theta\left[\phi(z)_i\,\phi(z)_i^T\right] = \mathbb{E}_\theta[\phi(z_i)\,\phi(z_i)^T] = \mathcal{T}(\mathcal{K}_z). \tag{3}$$

Since $\phi$ does not introduce any new variables $\Theta_y$ can be computed as,

$$\Theta_y = \mathbb{E}_\theta\left[\frac{\partial \phi(z_i)}{\partial \theta}\left(\frac{\partial \phi(z_i)}{\partial \theta}\right)^T\right] = \mathbb{E}_\theta\left[\text{diag}(\dot{\phi}(z_i))\frac{\partial z_i}{\partial \theta}\left(\frac{\partial z_i}{\partial \theta}\right)^T \text{diag}(\dot{\phi}(z_i))\right] = \dot{\mathcal{T}}(\mathcal{K}_z) \odot \Theta_z.$$

Taken together these equations imply that,

$$\left(\mathcal{K}_y, \Theta_y\right) = \phi^*\left(\mathcal{K}_z, \Theta_z\right) \equiv \left(\mathcal{T}(\mathcal{K}_z),\ \dot{\mathcal{T}}(\mathcal{K}_z) \odot \Theta_z\right) \tag{4}$$

---

[5]$\mathcal{T}\left(\Sigma\right) \equiv \mathbb{E}\left[\phi(u)\phi(u)^T\right], \dot{\mathcal{T}}\left(\Sigma\right) \equiv \mathbb{E}\left[\phi'(u)\phi'(u)^T\right], u \sim \mathcal{N}\left(0, \Sigma\right)$, as in (Lee et al., 2019).

will be the translation rule for a pointwise nonlinearity. Note that Equation Equation 4 only has an analytic expression for a small set of activation functions $\phi$.

Next we consider the case of a dense operation. Using the independence between the weights, the biases, and $h$ it follows that,

$$\mathcal{K}_h = \mathbb{E}_{W,\beta,\theta}[h_i h_i^T] = \sigma_\omega^2 \mathbb{E}_\theta[y_i y_i^T] + \sigma_b^2 = \sigma_\omega^2 \mathcal{K}_y + \sigma_b^2. \tag{5}$$

Finally, the NTK of $h$ can be computed as a sum of two terms:

$$\Theta_h = \mathbb{E}_{W,\beta,\theta}\left[\frac{\partial h_i}{\partial(W,\beta)}\left(\frac{\partial h_i}{\partial(W,\beta)}\right)^T\right] + \mathbb{E}_{W,\beta,\theta}\left[\frac{\partial h_i}{\partial \theta}\left(\frac{\partial h_i}{\partial \theta}\right)^T\right] = \sigma_\omega^2 \mathcal{K}_y + \sigma_b^2 + \sigma_\omega^2 \Theta_y. \tag{6}$$

This gives the translation rule for the dense layer in terms of $\mathcal{K}_y$ and $\Theta_y$ as,

$$(\mathcal{K}_h, \Theta_h) = \mathrm{Dense}(\sigma_\omega, \sigma_b)^*(\mathcal{K}_y, \Theta_y) \equiv \left(\sigma_\omega^2 \mathcal{K}_y + \sigma_b^2, \ \sigma_\omega^2 \mathcal{K}_y + \sigma_b^2 + \sigma_\omega^2 \Theta_y\right). \tag{7}$$

## 3.2 PERFORMANCE

Our library performs a number of automatic performance optimizations without sacrificing flexibility.

**Leveraging block-diagonal covariance structure.** A common computational challenge with GPs is inverting the training set covariance matrix. Naively, for a classification task with $C$ classes and training set $\mathcal{X}$, NNGP and NTK covariances have the shape of $|\mathcal{X}|\,C \times |\mathcal{X}|\,C$. For CIFAR-10, this would be $500,000 \times 500,000$. However, if a fully-connected readout layer is used (which is an extremely common design in classification architectures), the $C$ logits are i.i.d. conditioned on the input $x$. This results in outputs that are normally distributed with a block-diagonal covariance matrix of the form $\Sigma \otimes I_C$, where $\Sigma$ has shape $|\mathcal{X}| \times |\mathcal{X}|$ and $I_C$ is the $C \times C$ identity matrix. This reduces the computational complexity and storage in many common cases by an order of magnitude, which makes closed-form exact inference feasible in these cases.

**Automatically tracking only the smallest necessary subset of intermediary covariance entries.** For most architectures, especially convolutional, the main computational burden lies in *constructing* the covariance matrix (as opposed to inverting it). Specifically for a convolutional network of depth $l$, constructing the $|\mathcal{X}| \times |\mathcal{X}|$ output covariance matrix, $\Sigma$, involves computing $l$ intermediate layer covariance matrices, $\Sigma^l$, of size $|\mathcal{X}|\,d \times |\mathcal{X}|\,d$ (see Listing 1 for a model requiring this computation) where $d$ is the total number of pixels in the intermediate layer outputs (e.g. $d = 1024$ in the case of CIFAR-10 with `SAME` padding). However, as Xiao et al. (2018); Novak et al. (2019); Garriga-Alonso et al. (2019) remarked, if no pooling is used in the network the output covariance $\Sigma$ can be computed by only using the stack of $d$ $|\mathcal{X}| \times |\mathcal{X}|$-blocks of $\Sigma^l$, bringing the time and memory cost from $\mathcal{O}(|\mathcal{X}|^2 d^2)$ down to $\mathcal{O}(|\mathcal{X}|^2 d)$ per layer (see Figure 4 and Listing 2 for models admitting this optimization). Finally, if the network has no convolutional layers, the cost further reduces to $\mathcal{O}(|\mathcal{X}|^2)$ (see Listing 3 for an example). These choices are performed automatically by NEURAL TANGENTS to achieve efficient computation and minimal memory footprint.

**Expressing covariance computations as 2D convolutions with optimal layout.** A key insight to high performance in convolutional models is that the covariance propagation operator for convolutional layers $\mathcal{A}$ can be expressed in terms of 2D convolutions when it operates on both the full $|\mathcal{X}|\,d \times |\mathcal{X}|\,d$ covariance matrix $\Sigma$, and on the $d$ diagonal $|\mathcal{X}| \times |\mathcal{X}|$-blocks. This allows utilization of modern hardware accelerators, many of which target 2D convolutions as their primary machine learning application.

**Simultaneous NNGP and NT kernel computations.** As NTK computation requires the NNGP covariance as an intermediary computation, the NNGP covariance is computed together with the NTK at no extra cost. This is especially convenient for researchers looking to investigate similarities and differences between these two infinite-width NN limits.

**Automatic batching and parallelism across multiple devices.** In most cases as the dataset or model becomes large, it is impossible to perform the entire kernel computation at once. Additionally, in many cases it is desirable to parallelize the kernel computation across devices (CPUs, GPUs, or TPUs). NEURAL TANGENTS provides an easy way to perform both of these common tasks using a single `batch` decorator shown below:

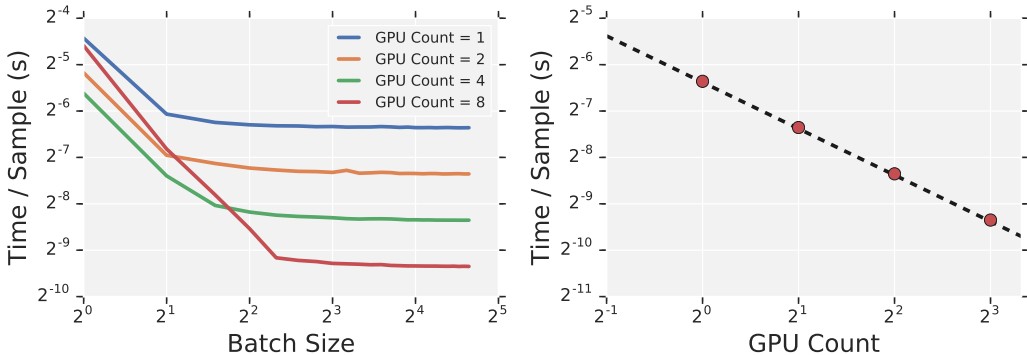

Figure 5: **Performance scaling with batch size (left) and number of GPUs (right).** Shows time per entry needed to compute the analytic NNGP and NTK covariance matrices (using `kernel_fn`) in a 21-layer ReLU network with global average pooling. **Left:** Increasing the batch size when computing the covariance matrix in blocks allows for a significant performance increase until a certain threshold when all cores in a single GPU are saturated. Simpler models are expected to have better scaling with batch size. **Right:** Time-per-sample scales linearly with the number of GPUs, demonstrating near-perfect hardware utilization.

```
batched_kernel_fn = nt.batch(kernel_fn, batch_size)
batched_kernel_fn(x, x) == kernel_fn(x, x)  # True!
```

This code works with either analytic kernels or empirical kernels. By default, it automatically shares the computation over all available devices. We plot the performance as a function of batch size and number of accelerators when computing the theoretical NTK of a 21-layer convolutional network in Figure 5, observing near-perfect scaling with the number of accelerators.

**Op fusion.** JAX and XLA allow end-to-end compilation of the whole kernel computation and/or inference. This enables the XLA compiler to fuse low-level ops into custom model-specific accelerator kernels, as well as eliminating overhead from op-by-op dispatch to an accelerator. In similar vein, we allow the covariance tensor to change its order of dimensions from layer to layer, with the order tracked and parsed as additional metadata under the hood. This eliminates redundant transpositions[6] by adjusting the computation performed by each layer based on the input metadata.

## 4 CONCLUSION

We believe NEURAL TANGENTS will enable researchers to quickly and easily explore infinite-width networks. By democratizing this previously challenging model family, we hope that researchers will begin to use infinite neural networks, in addition to their finite counterparts, when faced with a new problem domain (especially in cases that are data-limited). In addition, we are excited to see novel uses of infinite networks as theoretical tools to gain insight and clarity into many of the hard theoretical problems in deep learning. Going forward, there are significant additions to NEURAL TANGENTS that we are exploring. There are more layers we would like to add in the future (§D) that will enable an even larger range of infinite network topologies. Additionally, there are further performance improvements we would like to implement, to allow experimenting with larger models and datasets. We invite the community to join our efforts by contributing new layers to the library (§B.7), or by using it for research and providing feedback!

ACKNOWLEDGMENTS

We thank Yasaman Bahri for frequent discussion and useful feedback on the manuscript. We additionally appreciate both Yasaman Bahri and Greg Yang for the ongoing contributions to improve the library. We thank Sergey Ioffe for feedback on the text, as well as Ravid Ziv, and Jeffrey Pennington for discussion and feedback on early versions of the library.

---

[6]These transpositions could not be automatically fused by the XLA compliler.

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

## APPENDIX

### A  NEURAL TANGENTS  AND PRIOR WORK

Here we briefly discuss the differences between NEURAL TANGENTS and the relevant prior work.

1. **Prior benchmarks in the domain of infinitely wide neural networks.** Various prior works have evaluated convolutional and fully-connected models on certain datasets (Lee et al., 2018; Matthews et al., 2018b;a; Novak et al., 2019; Garriga-Alonso et al., 2019; Arora et al., 2019a). While these efforts must have required implementing certain parts of our library, to our knowledge such prior efforts were either not open-sourced or not comprehensive / user-friendly / scalable enough to be used as a user-facing library. In addition, all of the works above used their own separate implementation, which further highlights a need for a more general approach.

2. **Code released by Lee et al. (2019).** Lee et al. (2019) have released code along with their paper submission, which is a strict and minor subset of our library. More specifically, at the time of the submission, Lee et al. (2019) have released code equivalent to `nt.linearize`, `nt.empirical_ntk_fn`, `nt.predict.gradient_descent_mse`, `nt.predict.gradient_descent`, and `nt.predict.momentum`. Every other part of the library (most notably, `nt.stax`) is new in this submission and was not used by Lee et al. (2019) or any other prior work. At the time of writing, NEURAL TANGENTS differs from the code released by Lee et al. (2019) by about $+9,500/-2,500$ lines of code.

3. **GPy (2012), GPFlow (Matthews et al., 2017), GPyTorch (Gardner et al., 2018), and other GP packages.** While various packages allowing for kernel construction, optimization, and inference with Gaussian Processes exist, none of them allow easy construction of the very specific kernels corresponding to infinite neural networks (NNGP/NTK; `nt.stax`), nor do they provide the tools and convenience for studying wide but finite networks and their training dynamics (`nt.taylor_expand`, `nt.predict`, `nt.monte_carlo_kernel_fn`). On the other hand, NEURAL TANGENTS does not provide any tools for approximate inference with these kernels.

## B  LIBRARY DESCRIPTION

NEURAL TANGENTS provides a high-level interface for specifying analytic, infinite-width, Bayesian and gradient descent trained neural networks as Gaussian Processes. This interface closely follows the `stax` API (Bradbury et al., 2018b) in JAX.

### B.1  NEURAL NETWORKS WITH JAX

`stax` represents each component of a network as two functions: `init_fn` and `apply_fn`. These components can be composed in serial or in parallel to produce new network components with their own `init_fn` and `apply_fn`. In this way, complicated neural network architectures can be specified hierarchically.

Calling `init_fn` on a random seed and an input shape generates a random draw of trainable parameters for a neural network. Calling `apply_fn` on these parameters and a batch of inputs returns the outputs of the given finite neural network.

```python
from jax.experimental import stax
init_fn, apply_fn = stax.serial(stax.Dense(512), stax.Relu, stax.Dense(10))
_, params = init_fn(key, (-1, 32 * 32 * 3))
fx_train, fx_test = apply_fn(params, x_train),  apply_fn(params, x_test)
```

### B.2  INFINITE NEURAL NETWORKS WITH NEURAL TANGENTS

We extend `stax` layers to return a third function `kernel_fn`, which represents the covariance functions of the infinite NNGP and NTK networks of the given architecture (recall that since infinite networks are GPs, they are fully defined by their covariance functions, assuming 0 mean as is common in the literature).

```python
from neural_tangents import stax
init_fn, apply_fn, kernel_fn = stax.serial(stax.Dense(512), stax.Relu(), stax.Dense(10))
```

We demonstrate a specification of a more complicated architecture (WideResNet) in Listing 1.

`kernel_fn` accepts two batches of inputs `x1` and `x2` and returns their NNGP covariance and NTK matrices as `kernel_fn(x1, x2).nngp` and `kernel_fn(x1, x2).ntk` respectively, which

can then be used to make posterior test set predictions as the mean of a conditional multivariate normal $\mathcal{Y}_{\text{test}} = \mathcal{K}\left(\mathcal{X}_{\text{test}}, \mathcal{X}_{\text{train}}\right)\mathcal{K}\left(\mathcal{X}_{\text{train}}, \mathcal{X}_{\text{train}}\right)^{-1}\mathcal{Y}_{\text{train}}$:

```
from jax.numpy.linalg import inv
y_test = kernel_fn(x_test, x_train).ntk @ inv(kernel_fn(x_train, x_train).ntk) @ y_train
```

Note that the above code does not do Cholesky decomposition and is presented merely to show the mathematical expression. We provide efficient GP inference method in the `predict` submodule:

```
import neural_tangents as nt
y_test = nt.predict.gp_inference(kernel_fn, x_train, y_train, x_test,
                                 get='ntk', diag_reg=1e-4, compute_cov=False)
```

## B.3 COMPUTING INFINITE NETWORK KERNELS IN BATCHES AND IN PARALLEL

Naively, the `kernel_fn` will compute the whole kernel in a single call on one device. However, for large datasets or complicated architectures, it is often necessary to distribute the calculation in some way. To do this, we introduce a `batch` decorator that takes a `kernel_fn` and returns a new `kernel_fn` with the exact same signature. The new function computes the kernel in batches and automatically parallelizes the calculation over however many devices are available, with near-perfect speedup scaling with the number of devices (Figure 5, right).

```
import neural_tangents as nt
kernel_fn = nt.batch(kernel_fn, batch_size=32)
```

Note that batching is often used to compute large covariance matrices that may not even fit on a GPU/TPU device, and require to be stored and used for inference using CPU RAM. This is easy to achieve by simply specifying `nt.batch(..., store_on_device=False)`. Once the matrix is stored in RAM, inference will be performed with a CPU when `nt.predict` methods are called. As mentioned in §3.2, for many (notably, convolutional, and especially pooling) architectures, inference cost can be small relative to kernel construction, even when running on CPU (for example, it takes less than 3 minutes to execute `jax.scipy.linalg.solve(..., sym_pos=True)` on a $45,000 \times 45,000$ training covariance matrix and a $45,000 \times 10$ training target matrix).

## B.4 TRAINING DYNAMICS OF INFINITE NETWORKS

In addition to closed form multivariate Gaussian posterior prediction, it is also interesting to consider network predictions following continuous gradient descent. To facilitate this we provide several functions to compute predictions following gradient descent with an MSE loss, for gradient descent with arbitrary loss, or for momentum with arbitrary loss. The first case is handled analytically, while the latter two are computed by numerically integrating the differential equation. For example, the following code will compute the function evaluation on train and test points following gradient descent for some time `training_time`.

```
import neural_tangents as nt
predictor = nt.predict.gradient_descent_mse(kernel_fn(x_train, x_train), y_train,
fx_train, fx_test = predictor(training_time, fx_train, fx_test)
```

## B.5 INFINITE NETWORKS OF ANY ARCHITECTURE THROUGH SAMPLING

Of course, there are cases where the analytic kernel cannot be computed. To support these situations, we provide utility functions to efficiently compute Monte Carlo estimates of the NNGP covariance and NTK. These functions work with neural networks constructed using *any* neural network library.

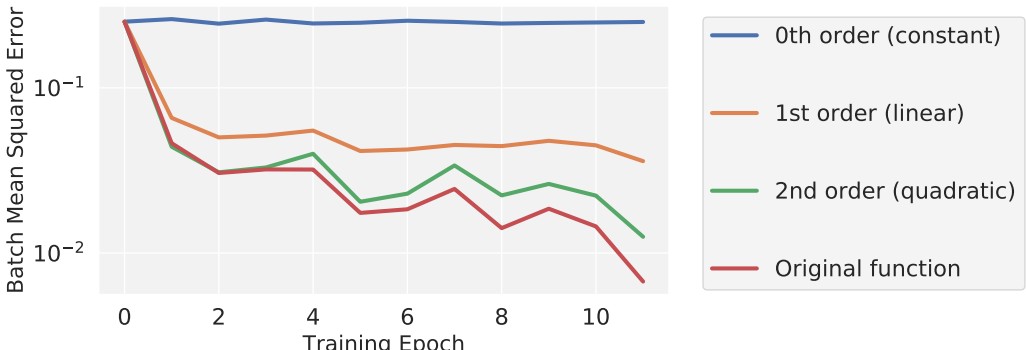

Figure 6: **Training a neural network and its various approximations using** `nt.taylor_expand`. Presented is a **5-layer Erf-neural network of width 512** trained on MNIST using SGD with momentum, along with its **constant (0th order)**, **linear (1st order)**, and **quadratic (2nd order)** Taylor expansions about the initial parameters. As training progresses (left to right), lower-order expansions deviate from the original function faster than higher-order ones.

```
from jax import random
from jax.experimental import stax
import neural_tangents as nt

init_fn, apply_fn = stax.serial(stax.Dense(64), stax.BatchNorm(), stax.Sigmoid, stax.Dense(1))
kernel_fn = nt.monte_carlo_kernel_fn(init_fn, apply_fn, key=random.PRNGKey(1), n_samples=128)
kernel = kernel_fn(x_train, x_train)
```

We demonstrate convergence of the Monte Carlo kernel estimates to the closed-form analytic kernels in the case of a WideResNet in Figure 2.

### B.6 WEIGHTS OF WIDE BUT FINITE NETWORKS

While most of NEURAL TANGENTS is devoted to a function-space perspective—describing the distribution of function values on finite collections of training and testing points—we also provide tools to investigate a dual weight space perspective described in Lee et al. (2019). Convergence of dynamics to NTK dynamics coincide with networks being described by a linear approximation about their initial set of parameters. We provide decorators `linearize` and `taylor_expand` to approximate functions to linear order and to arbitrary order respectively. Both functions take an `apply_fn` and returns a new `apply_fn` that computes the series approximation.

```
import neural_tangents as nt
taylor_apply_fn = nt.taylor_expand(apply_fn, params, order)
fx_train_approx = taylor_apply_fn(new_params, x_train)
```

These act exactly like normal JAX functions and, in particular, can be plugged into gradient descent, which we demonstrate in Figure 6.

### B.7 EXTENDING NEURAL TANGENTS

Many neural network layers admit a sensible infinite-width limit behavior in the Bayesian and continuous gradient descent regimes as long as the multivariate central limit theorem applies to their outputs conditioned on their inputs. To add such layer to NEURAL TANGENTS, one only has to implement it as a method in `nt.stax` with the following signature:

```
@_layer  # an internal decorator taking care of certain boilerplate.
NewLayer(layer_params: Any) -> (init_fn: function, apply_fn: function, kernel_fn: function)
```

Here `init_fn` and `apply_fn` are initialization and the forward pass methods of the finite width layer implementation (see §B.1). If the layer of interest already exists in JAX, there is no need to implement these methods and the user can simply return the respective methods from `jax.experimental.stax` (see `nt.stax.Flatten` for an example; in fact the majority of `nt.stax` layers call the original `jax.experimental.stax` layers for finite width layer methods).

In this case what remains is to implement the `kernel_fn` method with signature

```
kernel_fn(input_kernel: nt.utils.Kernel) -> output_kernel: nt.utils.Kernel
```

Here both `input_kernel` and `output_kernel` are `namedtuple`s containing the NNGP and NTK covariance matrices, as well as additional metadata useful for computing the kernel propagation operation. The specific operation to be performed should be derived by the user in the context of the particular operation that the finite width layer performs. This transformation could be as simple as an affine map on the kernel matrices, but could also be analytically intractable.

Once implemented, the correctness of the implementation can be very easily tested by extending the `nt.tests.stax_test` with the new layer, to test the agreement with large-widths empirical NNGP and NTK kernels.

## C   ARCHITECTURE SPECIFICATIONS

```python
from neural_tangents import stax

def ConvolutionalNetwork(depth, W_std=1.0, b_std=0.0):
  layers = []
  for _ in range(depth):
    layers += [stax.Conv(1, (3, 3), W_std, b_std, padding='SAME'), stax.Relu()]
  layers += [stax.Flatten(), stax.Dense(1, W_std, b_std)]
  return stax.serial(*layers)
```

Listing 2: **All-convolutional model (ConvOnly) definition used in Figure 3**.

```python
from neural_tangents import stax

def FullyConnectedNetwork(depth, W_std=1.0, b_std=0.0):
  layers = [stax.Flatten()]
  for _ in range(depth):
    layers += [stax.Dense(1, W_std, b_std), stax.Relu()]
  layers += [stax.Dense(1, W_std, b_std)]
  return stax.serial(*layers)
```

Listing 3: **Fully-connected (FC) model definition used in Figure 3**.

## D   IMPLEMENTED AND COMING SOON FUNCTIONALITY

The following layers[7] are currently implemented, with translation rules given in Table 1:

- `serial`
- `parallel`
- `FanOut`
- `FanInSum`
- `FanInConcat`
- `Dense`
- `Conv` with arbitrary filter shapes, strides, dimension numbers, and padding[8]
- `Relu`
- `LeakyRelu`
- `Abs`
- `ABRelu` [9]
- `Erf`
- `Identity`
- `Flatten`
- `AvgPool`
- `GlobalAvgPool`
- `SumPool`
- `GlobalSumPool`
- `Dropout`
- `LayerNorm`
- `GlobalSelfAttention` (Anonymous, 2020)

The following is in our near-term plans:

- `Exp` , `Elu` , `Selu` , `Gelu`
- Apache Beam support.

The following layers do *not* have known closed-form expressions for infinite network covariances, and respective infinite networks have to be estimated empirically (via `nt.monte_carlo_kernel_fn` ) or using other approximations (not currently implemented):

- `Sigmoid` , `Tanh` ,[10] `Swish` ,[11] `Softmax` , `LogSoftMax` , `Softplus` , `MaxPool` .

---

[7] `Abs` , `ABRelu` , `Erf` , `GlobalAvgPool` , `GlobalSumPool` , `LayerNorm` , and `GlobalSelfAttention` are only available in our library `nt.stax` and not in `jax.experimental.stax` .

[8]In addition to `SAME` and `VALID` , we support `CIRCULAR` padding, which is especially convenient for theoretical analysis and was used by Xiao et al. (2018) and Novak et al. (2019).

[9] `ABRelu` is a more flexible extension of `LeakyRelu` , computed as $a \min(x, 0) + b \max(x, 0)$.

[10] `Sigmoid` and `Tanh` are similar to `Erf` which does have a covariance expression and is implemented.

[11] `Swish` is similar to `Gelu` .

| Tensor Op | NNGP Op | NTK Op |
|---|---|---|
| $\mathcal{X}$ | $\mathcal{K}$ | $\Theta$ |
| Dense$(\sigma_w, \sigma_b)$ | $\sigma_w^2 \mathcal{K} + \sigma_b^2$ | $(\sigma_w^2 \mathcal{K} + \sigma_b^2) + \sigma_w^2 \Theta$ |
| $\phi$ | $\mathcal{T}(\mathcal{K})$ | $\dot{\mathcal{T}}(\mathcal{K}) \odot \Theta$ |
| Dropout$(\rho)$ | $\mathcal{K} + \left(\frac{1}{\rho} - 1\right) \mathrm{Diag}(\mathcal{K})$ | $\Theta + \left(\frac{1}{\rho} - 1\right) \mathrm{Diag}(\Theta)$ |
| Conv$(\sigma_w, \sigma_b)$ | $\sigma_w^2 \mathcal{A}(\mathcal{K}) + \sigma_b^2$ | $\sigma_w^2 \mathcal{A}(\mathcal{K}) + \sigma_b^2 + \sigma_w^2 \mathcal{A}(\Theta)$ |
| Flatten | $\mathrm{Tr}(\mathcal{K})$ | $\mathrm{Tr}(\mathcal{K} + \Theta)$ |
| AvgPool$(s, q, p)$ | AvgPool$(s, q, p)(\mathcal{K})$ | AvgPool$(s, q, p)(\mathcal{K} + \Theta)$ |
| GlobalAvgPool | GlobalAvgPool$(\mathcal{K})$ | GlobalAvgPool$(\mathcal{K} + \Theta)$ |
| SumPool$(s, q, p)$ | SumPool$(s, q, p)(\mathcal{K})$ | SumPool$(s, q, p)(\mathcal{K} + \Theta)$ |
| GlobalSumPool | GlobalSumPool$(\mathcal{K})$ | GlobalSumPool$(\mathcal{K} + \Theta)$ |
| Attn$(\sigma_{QK}, \sigma_{OV})$ (Anonymous, 2020) | Attn$(\sigma_{QK}, \sigma_{OV})(\mathcal{K})$ | 2Attn$(\sigma_{QK}, \sigma_{OV})(\mathcal{K})+$ Attn$(\sigma_{QK}, \sigma_{OV})(\Theta)$ |
| FanInSum$(\mathcal{X}_1, \dots, \mathcal{X}_n)$ | $\sum_{j=1}^n \mathcal{K}_j$ | $\sum_{j=1}^n \Theta_j$ |
| FanOut$(n)$ | $[\mathcal{K}] * n$ | $[\Theta] * n$ |

Table 1: **Translation rules (§3) converting tensor operations into operations on NNGP and NTK kernels.** Here the input tensor $\mathcal{X}$ is assumed to have shape $|\mathcal{X}| \times H \times W \times C$ (dataset size, height, width, number of channels), and the full NNGP and NT kernels $\mathcal{K}$ and $\mathcal{T}$ are considered to be of shape $(|\mathcal{X}| \times H \times W)^{\times 2}$ (in practice shapes of $|\mathcal{X}|^{\times 2} \times H \times W$ and $|\mathcal{X}|^{\times 2}$ are also possible, depending on which optimizations in §3.2 are applicable). **Notation details.** The Tr, GlobalAvgPool, and GlobalSumPool ops are assumed to act on all spatial axes (with sizes $H$ and $W$ in this example), producing a $|\mathcal{X}|^{\times 2}$-kernel. Similarly, the AvgPool and SumPool ops is assumed to act on all spatial axes as well, applying the specified strides $s$, pooling window sizes $p$ and padding strategy $p$ to the respective axes pairs in $\mathcal{K}$ and $\mathcal{T}$ (acting as 4D pooling with replicated parameters of the 2D version). $\mathcal{T}$ and $\dot{\mathcal{T}}$ are defined identically to Lee et al. (2019) as $\mathcal{T}(\Sigma) = \mathbb{E}\left[\phi(u)\phi(u)^T\right], \dot{\mathcal{T}}(\Sigma) = \mathbb{E}\left[\phi'(u)\phi'(u)^T\right], u \sim \mathcal{N}(0, \Sigma)$. These expressions can be evaluated in closed form for many nonlinearities, and preserve the shape of the kernel. The $\mathcal{A}$ op is defined similarly to Novak et al. (2019); Xiao et al. (2018) as $[\mathcal{A}(\Sigma)]_{h,h'}^{w,w'}(x, x') = \sum_{dh,dw}[\Sigma]_{h+dh,h'+dh}^{w+dw,w'+dw}(x, x')/q^2$, where the summation is performed over the convolutional filter receptive field with $q$ pixels (we assume unit strides and circular padding in this expression, but generalization to other settings is trivial and supported by the library). $[\Sigma] * n = [\Sigma, \dots, \Sigma]$ ($n$-fold replication). For LayerNorm, FanInConcat, and Attn (Anonymous, 2020) translation rules we refer the reader to our code at https://github.com/google/neural-tangents, as these ops are challenging to express concisely using current notation. See Figure 4 for an example of applying the translation rules to a specific model, and §3.1 for deriving a sample translation rule. See §D for the full list of currently implemented translations.

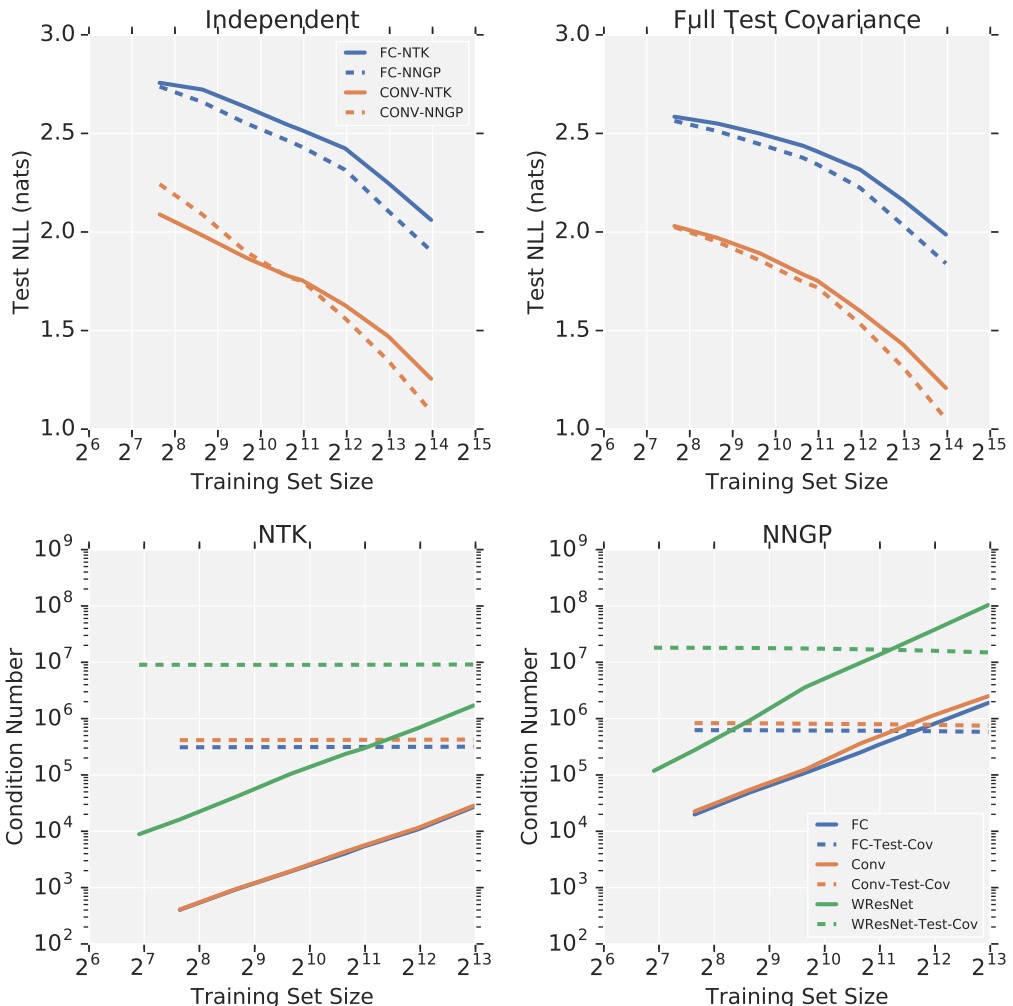

Figure 7: **Predictive negative log-likelihoods and condition numbers. Top.** Test negative log-likelihoods for NNGP posterior and Gaussian predictive distribution for NTK at infinite training time for CIFAR-10 (test set of 2000 points). **Fully Connected (FC, Listing 3)** and **Convolutional network without pooling (CONV, Listing 2)** models are selected based on train marginal negative log-likelihoods in Figure 3. **Bottom.** Condition numbers for covariance matrices corresponding to NTK/NNGP as well as respective predictive covaraince on the test set. Ill-conditioning of **Wide Residual Network** kernels due to pooling layers (Xiao et al., 2019) could be the cause of numerical issues when evaluating predictive NLL for this kernels.

