# OpenReview forum: "Neural Tangents: Fast and Easy Infinite Neural Networks in Python"
_ICLR.cc/2020/Conference — Accept (Spotlight)_

### Official Review · AnonReviewer2 · 2019-10-23
**Official Blind Review #2**

**Rating:** 6

**Review:**

This work develops a library for working with a class of infinitely wide neural networks, in particular those corresponding to neural tangent kernels (NTKs) and neural network Gaussian processes (NNGPs). The theory for these two kernels was well developed in a series of recent papers, and this library provides an automatic way to transform any appropriate neural net architecture into its corresponding NTK and NNGP.

Infinitely wide neural networks have been a popular subject of theoretical research and have been observed to have highly nontrivial performance on a variety of tasks (e.g. CIFAR-10 classification). It's really nice to see the development of such a library, which I believe could benefit the deep learning community a lot, especially for theoretical research on NTK.

I appreciate this work a lot. Currently I can only give weak accept instead of accept for a couple of reasons:
1. The theory and formulae of NTKs and NNGPs were well developed. This work mostly consists of implementing and modularizing them. The research contribution is relatively low.
2. As commented in the paper and if I understand correctly, the current library cannot scale to large datasets for CNNs with pooling. This would make the computation much more expensive (and probably infeasible without additional techniques and huge computing power) as mentioned in [Novak et al. 2019] and [Arora et al. 2019]. However pooling seems extremely useful for NTKs and NNGPs on image datasets. I think this makes this work somewhat less exciting than it may sound.

**Experience Assessment:**

I have published one or two papers in this area.

**Review Assessment: Checking Correctness Of Derivations And Theory:**

I assessed the sensibility of the derivations and theory.

**Review Assessment: Checking Correctness Of Experiments:**

I assessed the sensibility of the experiments.

**Review Assessment: Thoroughness In Paper Reading:**

I read the paper thoroughly.

---

> ### Author Response · Authors · 2019-11-15
> **AnonReviewer2 Rebuttal**
>
> Thank you for the careful review, we’re happy you found our library useful and beneficial to the ML community. Below we believe we have addressed your concerns, and we hope you can increase your score as a result.
>
>
> ----------------------------------------------------------------------------------------------------
> >>> 1. The theory and formulae of NTKs and NNGPs were well developed. This work mostly consists of implementing and modularizing them. The research contribution is relatively low.
>
> ICLR explicitly calls for “implementation issues, parallelization, software platforms, hardware” (https://iclr.cc/, bottom). We believe that Neural Tangents will unlock qualitatively new avenues for research by making computations on infinite networks tractable for non-experts and orders of magnitude easier for theoretical practitioners. This is increasingly true as work on infinite networks continues to attract interest from the community.
>
> On a separate note, a significant intellectual effort went into designing and efficiently implementing the library while keeping it scalable, flexible, and easy to use (see section 3.2 [new revision number]). By means of analogy, there is an immense gap between knowing the mathematical formulae of a convolutional layer and having a general and user-/accelerator-friendly implementation. We believe this kind of gaps should not be underestimated, and the novelty of our approach is itself a research contribution.
>
>
> ----------------------------------------------------------------------------------------------------
> >>> 2. As commented in the paper and if I understand correctly, the current library cannot scale to large datasets for CNNs with pooling. This would make the computation much more expensive (and probably infeasible without additional techniques and huge computing power) as mentioned in [Novak et al. 2019] and [Arora et al. 2019]. However pooling seems extremely useful for NTKs and NNGPs on image datasets. I think this makes this work somewhat less exciting than it may sound.
>
> You are correct that CNN-GPs/NTKs with pooling are _very_ compute-hungry. However, we would like to highlight that
>
> 1) We did successfully run experiments on 8K CIFAR10 subsets for a WideResNet with pooling in Figure 3, and we have further run pooling experiments on the 45K CIFAR10 training set and achieved a slight improvement over the prior state of the art in [Arora et al. 2019a] with our library (see the table below, GAP = global average pooling, best values marked with **).
>
> ╔══════════════════════╦═══════╦══════╦════════╦═══════╗
> ║ Model                                               ║ NNGP acc ║ NTK acc ║ NNGP loss  ║ NTK loss   ║
> ╠══════════════════════╬═══════╬══════╬════════╬═══════╣
> ║ WResNet-LayerNorm-depth_28   ║ 73.7           ║ 72.8        ║ 0.0501         ║ 0.0501      ║
> ╠══════════════════════╬═══════╬══════╬════════╬═══════╣
> ║ CNN-GAP-Relu-depth_10               ║ 78.84        ║ *77.84*  ║ 0.0454         ║ *0.0462*  ║
> ╠══════════════════════╬═══════╬══════╬════════╬═══════╣
> ║ CNN-GAP-Relu-depth_20               ║ *79.38*    ║ 76.98      ║ *0.0447*    ║ *0.0462*  ║
> ╠══════════════════════╬═══════╬══════╬════════╬═══════╣
> ║ CNN-GAP-Erf-depth_10                  ║ 71.32        ║ 71.3        ║ 0.0538         ║ 0.054         ║
> ╠══════════════════════╬═══════╬══════╬════════╬═══════╣
> ║ Arora et al. 2019 [GAP]                  ║ -                 ║ 77.43      ║ -                   ║ -                 ║
> ╠══════════════════════╬═══════╬══════╬════════╬═══════╣
> ║ Li et al. 2019b [GAP]                       ║ 78.49         ║ 77.63      ║ -                   ║ -                 ║
> ╚══════════════════════╩═══════╩══════╩════════╩═══════╝
>
> [Arora et al. 2019] https://arxiv.org/pdf/1904.11955v2.pdf
> [Li et al. 2019] https://arxiv.org/pdf/1911.00809v1.pdf
>
>
> 2) Our library provides the [parallelizable] "nt.monte_carlo_fn" method to Monte-Carlo estimate compute-heavy kernels, and we have established reasonable convergence for a WideResNet in Figure 2. The question of how good of a tradeoff between accuracy and time / memory the MC method provides is still admittedly open and left for future work.
>
> 3) Pooling CNN kernels is arguably an emerging field of study, and we believe that as groups with large computing power demonstrate their good performance, studying these kernels (e.g. on small datasets) and developing novel approximation / mimicking / “inspired-by” techniques will attract a lot of research attention. We believe our library will facilitate such research greatly, and serve as a platform to deliver new results to the users.

---

### Official Review · AnonReviewer1 · 2019-10-23
**Official Blind Review #1**

**Rating:** 8

**Review:**

POST-REBUTTAL COMMENTS

I appreciate the response from the authors.

I particularly like the comparison table in the response to the other reviewer and ought to be highlighted in the paper.

If I were to start this line of research, I would be inclined to expand on the codebase. The contribution is significant. Hence, I am bumping up my score to accept.


PRIOR FEEDBACK

The contribution of this work lies in providing a library for working with the existing variants of infinite-width neural networks and avoiding the need to derive the NNGP and NT kernels for each architecture by hand. The authors have firstly shown performance comparisons between inferences between finite vs. infinitely wide neural networks. The authors then go into some implementation details with their library. The authors have provided the code and cookbook in the links provided in the abstract. On the overall, I like this effort which is timely.

Some additional suggestions below:

I would like to see an additional metric for performance comparison of probabilistic models, which is often used in the GP literature: mean negative log probability.

It would also be interesting to see how the posterior variance (e.g., Fig. 1 right) evolves over the entire space during training.

I would have preferred a more detailed discussion about the implementation on transforming tensor ops to kernel ops in Section 3.

For the summary of contributions, can you give the corresponding section number to refer to when you demonstrate each feature? For example, is the 4th feature (i.e., exploring weight space perspective) demonstrated in the paper?

Can the authors elaborate on the ease of expanding their library for the new developments in this field?


Minor issues:

Page 1: Gaussian Procesesses?
Page 4: it’s infinite?
Fig. 4: I would have preferred the indices to be placed as subscripts instead of superscripts.
Page 8: it’s order of dimensions?

**Experience Assessment:**

I have read many papers in this area.

**Review Assessment: Checking Correctness Of Derivations And Theory:**

I carefully checked the derivations and theory.

**Review Assessment: Checking Correctness Of Experiments:**

I carefully checked the experiments.

**Review Assessment: Thoroughness In Paper Reading:**

I read the paper thoroughly.

---

> ### Author Response · Authors · 2019-11-15
> **Suggestions implemented!**
>
> Thank you for the careful review and great suggestions! We believe to have addressed all your comments, and hope that you could increase the score as a result.
>
>
> --------------------------------------------------------------------------------------------------
> >>> I would like to see an additional metric for performance comparison of probabilistic models, which is often used in the GP literature: mean negative log probability.
>
> Thank you for the suggestion, we have added negative log likelihood (NLL) measurement in the updated version. With model’s marginal likelihood (train) we did model selection across different depths and plotted accuracy/mean squared error/marginal NLL. In the appendix (Figure 7), we included test NLL’s for fully connected and convolutional models. Since the predictive covariance of the WideResNet kernel has high condition number (due to the pooling layers, see https://openreview.net/pdf?id=Bkx1mxSKvB section C), obtaining numerical stable NLL measures was more challenging.
>
>
> --------------------------------------------------------------------------------------------------
> >>> It would also be interesting to see how the posterior variance (e.g., Fig. 1 right) evolves over the entire space during training.
>
> Thank you for the suggestion, done in the new revision!
>
>
> --------------------------------------------------------------------------------------------------
> >>> I would have preferred a more detailed discussion about the implementation on transforming tensor ops to kernel ops in Section 3.
>
> Agreed - in the new revision, we have expanded the text with section 3.1. demonstrating the tensor-to-kernel ops translation.
>
>
> --------------------------------------------------------------------------------------------------
> >>> For the summary of contributions, can you give the corresponding section number to refer to when you demonstrate each feature? For example, is the 4th feature (i.e., exploring weight space perspective) demonstrated in the paper?
>
> Great suggestion, done in the new revision. We have also added an experiment demonstrating linearization / taylor expansion (4th feature, section B.6, Figure 6). Please also see the existing example in `examples/weight_space.py` and https://github.com/neural-tangents/neural-tangents#weight-space.
>
>
> --------------------------------------------------------------------------------------------------
> >>> Can the authors elaborate on the ease of expanding their library for the new developments in this field?
>
> Thank you for the question, we have elaborated on the process of extending the library to new layers in the new revision in section B.7 (see also new section 3.1 for the mathematical aspect of deriving new NTK/NNGP results). In general, we believe the process to be fairly straightforward, apart from the cases of:
>
> - Certain nonlinearities: to derive the layer kernel propagation expression, the user has to compute the covariance of the nonlinearity (and its derivative, for NTK) applied to correlated Gaussian variables. As discussed in section E (new revision), some such nonlinearities may not have known exact expressions for these covariances, and either "nt.empirical_kernel_fn" or other specialized approximations need to be employed.
>
> - Weight sharing between different layers in the network is not currently supported and may require some nontrivial work, but it is on our radar.
>
> Finally, once we de-anonymize the repository, we will be using the Github issue and project tracker to inform and engage the community in the library development and planning, and provide support for users and developers!
>
>
> -------------------------------------------------------------------------------------------------
> >>> Minor issues:
>
> >>> Page 1: Gaussian Procesesses?
> >>> Page 4: it’s infinite?
> >>> Fig. 4: I would have preferred the indices to be placed as subscripts instead of superscripts.
> >>> Page 8: it’s order of dimensions?
>
> Thank you, all fixed in the new revision except for the Figure indices: we stick to superscript usage to follow an established tradition in prior work [1-4, ...] of using superscript for layer numbers and subscript for hidden units / channels.
>
> [1] Jaehoon Lee, Yasaman Bahri, Roman Novak, Sam Schoenholz, Jeffrey Pennington, and JaschaSohl-dickstein.  Deep neural networks as gaussian processes. https://arxiv.org/pdf/1711.00165.pdf
>
> [2] Alexander G. de G. Matthews, Jiri Hron, Mark Rowland, Richard E. Turner, and Zoubin Ghahramani.Gaussian process behaviour in wide deep neural networks. https://arxiv.org/pdf/1804.11271.pdf
>
> [3] Roman Novak, Lechao Xiao, Jaehoon Lee, Yasaman Bahri, Greg Yang, Jiri Hron, Daniel A. Abolafia,Jeffrey Pennington, and Jascha Sohl-Dickstein. Bayesian deep convolutional networks with many channels are gaussian processes. https://arxiv.org/pdf/1810.05148
>
> [4] Adrià Garriga-Alonso, Carl Edward Rasmussen, and Laurence Aitchison. Deep convolutional networks as shallow gaussian processes. https://arxiv.org/pdf/1808.05587.pdf

---

### Official Review · AnonReviewer3 · 2019-10-23
**Official Blind Review #3**

**Rating:** 3

**Review:**

Summary: A Jax based neural tangents kernel library is introduced, with native GPU, TPU, and XLA support. Due to the correspondences with infinite neural network kernels (NNGPs), these kernels are also able to be computed for (essentially) free. Layers which do not emit an analytical form (e.g. Tanh or Softplus) can be implemented using Monte Carlo estimates. Several engineering-based experiments are performed demonstrating the potential scalability of their library.

While I really enjoyed reading the paper and believe that this library could be extremely practically useful, I vote to reject this paper because I do not feel that it has sufficient novelty to be a paper on its own in light of Lee et al, 2019.

Edit: post rebuttal, I'm bumping my score to a weak reject but would have minimal qualms if this paper were to be accepted. I find the further experiments performed by the authors of very good quality overall, but I'm still not particularly satisfied by their `argument that the codebase itself is distinct enough from separate related work. It's unfortunately a bit hard, from a machine learning researcher side, to review the quality of a codebase in and of itself.

Significance: Having played around with the code a bit, I find that the library itself is of very high quality and is pretty straightforward to use. I could definitely see myself using this library in the future for research work.

However, my primary concern with this paper is that it’s not sufficiently distinct from the previous work of Lee et al, 2019. After all, most of the experiments in that paper would have required the type of implementation that is described in greater detail in this paper.

To be able to vote to accept this paper, I will have to see an experiment that is practically performed with the current library in order to distinguish it from previous work (specifically Lee et al, 2019). In recent work, Arora et al, 2019 (Note: I do not consider this reference in my review other than to be mentioned as an example of an experiment that could be run with your library) run neural tangent kernels on tabular data using kernel SVMs. One other potential example would be a kernel SVM in this manner on CIFAR-10. An alternative example would be to exploit the Gaussian process representation and test out both NTKs and NNGPs in comparison to standard kernels for GPs and NNs on UCI regression tasks.

Originality: Again, a very efficient and easy to use implementation of neural tangent kernels would be a great boost to the community. This is doubly so as Jax is easy and pretty straightforward to use and is quite numpy like.

Again, I am very concerned with originality in comparison to Lee et al, 2019. Even checking out the link to their codebase provides a github repo that is quite similar to this one. Given that ICLR is a venue of similar domain to NeurIPS, it’s not clear to me why this paper ought to be anything other than a separate supporting tech report. If this paper had been submitted to something like SysML (edit: or JMLR MLOSS), I would see the distinctness instead.

Clarity: I find the paper to be extremely well-written and easy to follow. The addition of code snippets throughout is very well done, even if it’s a bit overkill. I don’t know what adding a half page long description of an infinitely wide WideResNet adds to the paper when that space could be better used by another experiment.

Quality: I find the experiments performed to be very well constructed. Below are a few mostly minor comments on the experiments:

In Figure 1 on the right, I would have liked to have seen the posterior predictive for a NNGP with the same kernel as well.

In Figure 2, why is the NNGP slower to converge to the analytic values here? Obviously, the rates of convergence are the same, but the constants seem different.

In Figure 3 (and throughout the experiments), does “full Bayesian inference for CIFAR10” mean that you treated the classification labels as regression targets? If so, how was classification error measured.

In Section 3.1, you mention that the library “leverages block-diagonal structure” to invert the CIFAR10 covariance matrix for each class (still 50k x 50k). Possibly this is because I haven’t had the chance to use TPUs, but I’m currently struggling to see how one could form and invert (via Choleskys) matrices of this size (50k x 50k) on a standard GPU (or CPU). Could the authors please clarify how they did this (whether through iterative methods, another structure exploiting trick, lots of memory, etc.)?

second edit: I was also unable to respond to the final comment about the UCI experiments in its own comment, but thank you for providing the estimated depths. These results definitely show the potential software promise of the codebase and open some interesting new research questions as a result.

References:

Arora, S., et al., Harnessing the Power of Infinitely Wide Deep Nets on Small-data Tasks, https://arxiv.org/abs/1910.01663

Lee, J., et al., Wide Neural Networks of any Depth Evolve as Linear Models Under Gradient Descent, NeurIPS, 2019, https://arxiv.org/abs/1902.06720


**Experience Assessment:**

I have read many papers in this area.

**Review Assessment: Checking Correctness Of Derivations And Theory:**

I assessed the sensibility of the derivations and theory.

**Review Assessment: Checking Correctness Of Experiments:**

I carefully checked the experiments.

**Review Assessment: Thoroughness In Paper Reading:**

I read the paper at least twice and used my best judgement in assessing the paper.

---

> ### Author Response · Authors · 2019-11-15
> **[4/4] Summary Diff Table**
>
> For your convenience, we provide in this comment the diff
> https://github.com/neural-tangents/neural-tangents/compare/Lee_et_al_2019..master
>
> and the brief table of differences between the code released by Lee et al., 2019 at the time of their submission, and our work.
>
> Thank you again for the careful review. We hope, having addressed your concerns regarding differences between this work and Lee et al. that you will consider increasing your score.
>
> ╔════════════════════╦════════════════════╦══════════════════════════╗
> ║ Codebase                                    ║ Released with Lee et al., 2019 ║ Ours                                                             ║
> ╠════════════════════╬════════════════════╬══════════════════════════╣
> ║ Line of code                                ║ 1400+                                          ║ 6600+                                                            ║
> ╠════════════════════╬════════════════════╬══════════════════════════╣
> ║ Empirical kernel                        ║ NTK                                               ║ NTK/NNGP                                                  ║
> ╠════════════════════╬════════════════════╬══════════════════════════╣
> ║ Weight space linearization      ║ Yes                                                ║ Yes                                                               ║
> ╠════════════════════╬════════════════════╬══════════════════════════╣
> ║ Higher-order Taylor series expansion║ No                                   ║ Yes                                                                ║
> ╠════════════════════╬════════════════════╬══════════════════════════╣
> ║ Monte-carlo sampling for empirical NTK/NNGP ║ No                ║ Yes                                                                ║
> ╠════════════════════╬════════════════════╬══════════════════════════╣
> ║ Multi-device parallelization     ║ No                                                ║ Yes                                                                ║
> ╠════════════════════╬════════════════════╬══════════════════════════╣
> ║ Dense Layer                               ║ No                                                 ║ Yes                                                               ║
> ╠════════════════════╬════════════════════╬══════════════════════════╣
> ║ Nonlinearities                            ║ No                                                ║ ReLU, Erf, Abs, LeakyRelu, ABReLU         ║
> ╠════════════════════╬════════════════════╬══════════════════════════╣
> ║ Convolution                                ║ No                                                ║ Any paddings, strides, filter shapes       ║
> ╠════════════════════╬════════════════════╬══════════════════════════╣
> ║ Average pooling                        ║ No                                                ║ Global, local, any strides / shapes          ║
> ╠════════════════════╬════════════════════╬══════════════════════════╣
> ║ Flattening                                   ║ No                                                 ║ Yes                                                               ║
> ╠════════════════════╬════════════════════╬══════════════════════════╣
> ║ LayerNorm                                 ║ No                                                 ║ Yes                                                               ║
> ╠════════════════════╬════════════════════╬══════════════════════════╣
> ║ Skip-connection                         ║ No                                                ║ Yes                                                                ║
> ╠════════════════════╬════════════════════╬══════════════════════════╣
> ║ Global self-attention                 ║ No                                                ║ Yes                                                                ║
> ╠════════════════════╬════════════════════╬══════════════════════════╣
> ║ Finite-time inference of the posterior ║ NTK, Mean                    ║ Mean, covariance, NNGP/NTK                ║
> ╠════════════════════╬════════════════════╬══════════════════════════╣
> ║ Infinite-time inference of the posterior ║ No                               ║ Mean, covariance, NNGP/NTK                 ║
> ╠════════════════════╬════════════════════╬══════════════════════════╣
> ║ Dropout                                      ║ No                                                 ║ Coming soon                                              ║
> ╠════════════════════╬════════════════════╬══════════════════════════╣
> ║ Standard (non-ntk) parameterization ║ No                                  ║ Coming soon                                              ║
> ╚═════════════════════════╩═══════════════╩══════════════════════════╝

---

> ### Author Response · Authors · 2019-11-15
> **[3/4] Addressing Minor Comments**
>
> ----------------------------------------------------------------------------------------------------
> >>> In Figure 1 on the right, I would have liked to have seen the posterior predictive for a NNGP with the same kernel as well.
>
> Great idea, done in the new revision (Figure 1, left in red).
>
>
> ----------------------------------------------------------------------------------------------------
> >>> In Figure 2, why is the NNGP slower to converge to the analytic values here? Obviously, the rates of convergence are the same, but the constants seem different.
>
> Currently we are not aware of any rigorous results explaining the respective rates or constants. A very naive take is that empirical NTK, as an outer product of Jacobians, sums over a larger number of [admittedly dependent] random entries ("O(N^2 * d)", where "N" is width and "d" is depth) than NNGP, which is an outer product of the activations ("O(N)"). However, since the same random variables are involved in the computation of the NTK and NNGP, we are not certain the observed effect is not architecture / dataset dependent.
>
>
> ----------------------------------------------------------------------------------------------------
> >>> In Figure 3 (and throughout the experiments), does “full Bayesian inference for CIFAR10” mean that you treated the classification labels as regression targets? If so, how was classification error measured.
>
> You are correct, the targets were converted to mean-zero vectors like "[-0.1, …, 0.9, …, -0.1]", where 0.9 is assigned to the correct class index, and -0.1 to all others. The error was computed as the "1-accuracy" where accuracy is the fraction of samples where the argmax of the model output is at the correct class index. We have updated the description of Figure 3 in the new revision.
>
>
> ----------------------------------------------------------------------------------------------------
> >>> In Section 3.1, you mention that the library “leverages block-diagonal structure” to invert the CIFAR10 covariance matrix for each class (still 50k x 50k). Possibly this is because I haven’t had the chance to use TPUs, but I’m currently struggling to see how one could form and invert (via Choleskys) matrices of this size (50k x 50k) on a standard GPU (or CPU). Could the authors please clarify how they did this (whether through iterative methods, another structure exploiting trick, lots of memory, etc.)?
>
> Thank you for the question,
>
> 1) A 32-bit 50k x 50k matrix has a size of ~10Gb, which (together with auxiliary variables like targets and train-test kernel matrix) is pushing the limit of many modern GPUs/TPUs and inference is indeed not feasible on these accelerators.
>
> 2) However, calling "[jax.]scipy.linalg.solve(..., sym_pos=True)" is perfectly doable on a CPU, and runs in about 3 minutes on a laptop with 2.9 Ghz Intel Core i9 (6 cores) and 32 Gb of RAM for a 50k x 50k training set kernel matrix and 50k x 10 training targets. [P.S. due to a technical bug in JAX/XLA (https://github.com/google/jax/issues/1644) at the time of writing "jax.scipy.linalg.solve" fails for matrices larger than 46,340 x 46,340, but the issue is unrelated to compute/memory and the original "scipy.linalg.solve" works fine for 50k.]
>
> 3) Our library allows to effortlessly leverage both fast GPUs and typically larger amount of CPU RAM by computing the kernel on [multiple] GPUs and performing inference on CPU. For this the user only needs to pass "store_on_device=False" to the "batch" decorator (https://github.com/neural-tangents/neural-tangents/blob/408c07d938458bbe80da3e66e420eb1fb84cbe33/neural_tangents/utils/batch.py#L395), i.e. making the kernel be computed in batches on [multiple] GPUs and collected into a single matrix for further inference in the CPU RAM.
>
> We have expanded the discussion in the new revision (section B.3) to mention the above.

---

> > ### Comment · AnonReviewer3 · 2019-11-15
> > **Thank you for the revisions**
> >
> > Will number these for clarity:
> >
> > 1) Thanks for the updates on the comparison to the fully bayesian setting, I appreciate this comparison.
> >
> > 2) Thanks for specifically spelling out what's being done here - it makes the paper considerably more legible.
> >
> > 3) I had thought that might be the case - thank you for being specific as to how these matrices are being inverted.

---

> ### Author Response · Authors · 2019-11-15
> **[2/4] Addressing Major Comments**
>
> ----------------------------------------------------------------------------------------------------
> >>> To be able to vote to accept this paper, I will have to see an experiment that is practically performed with the current library in order to distinguish it from previous work (specifically Lee et al, 2019). In recent work, Arora et al, 2019 (Note: I do not consider this reference in my review other than to be mentioned as an example of an experiment that could be run with your library) run neural tangent kernels on tabular data using kernel SVMs. One other potential example would be a kernel SVM in this manner on CIFAR-10. An alternative example would be to exploit the Gaussian process representation and test out both NTKs and NNGPs in comparison to standard kernels for GPs and NNs on UCI regression tasks.
>
> Please note that _almost all_ the experiments in our paper used features specific to our library only. Precisely:
>
> - Figures 2, 3 require computing exact kernels for the infinitely WideResNet.
> - Figure 5 demonstrates scalability to multi-gpu machines.
> - All code listings demonstrate the main feature of our library which is seamless definition of any neural network architecture of both finite and infinite widths at no extra mental/typing cost.
> - Figure 6 (new revision) uses higher-order taylor expansion of a neural network.
>
> We stress that _none_ of the above were open-sourced or used in / necessary for Lee et al, 2019 [P.S. on a minor note, Figure 1 used the analytic Erf kernel, derived but not released / used in Lee et al, 2019]
>
> All this having been said, we agree that it would be nice to have a practical demonstration of the convenience provided by Neural Tangents on the example of Arora et al.’s results on the UCI dataset, as you suggested. Arora et al. provided clean code to reproduce their experiments. We seamlessly substituted the NTK implementation of Arora et al. with Neural Tangents. As a result, we were able to consider a wider range of architectures, finding that by selecting models on a per-experiment basis we were able to provide a marginal improvement of the Arora et al. result from 81.95% to 82.03%. We include a discussion of this experiment in Appendix C.
>
>
> ----------------------------------------------------------------------------------------------------
> [...] >>> Again, I am very concerned with originality in comparison to Lee et al, 2019. Even checking out the link to their codebase provides a github repo that is quite similar to this one. Given that ICLR is a venue of similar domain to NeurIPS, it’s not clear to me why this paper ought to be anything other than a separate supporting tech report. If this paper had been submitted to something like SysML (edit: or JMLR MLOSS), I would see the distinctness instead.
>
> Please see replies above. TL;DR code released with the submission of Lee et al. 2019 had a tiny fraction of the functionality our library offers, and (from personal correspondence) their paper neither used nor had to use the main features of our library (flexible, general, and efficient specifications and evaluation of exact NNGP/NTK kernels). We again stress that this can be verified by the diff link mentioned above (https://github.com/neural-tangents/neural-tangents/compare/Lee_et_al_2019..master), and we are happy to clarify any other questions regarding the overlap.
>
>
> ----------------------------------------------------------------------------------------------------
> >>> Clarity: I find the paper to be extremely well-written and easy to follow. The addition of code snippets throughout is very well done, even if it’s a bit overkill. I don’t know what adding a half page long description of an infinitely wide WideResNet adds to the paper when that space could be better used by another experiment.
>
> We have decided to highlight the WideResNet snippet since it:
>
> 1) Presents exactly the use-case that would be extremely tedious / not practical at all to implement without our library. Our library handles all the topology, striding, padding, performance optimizations etc, while allowing to specify the infinite networks simultaneously with the finite model, at _absolutely no_ extra mental effort.
>
> 2) Gives the reader a non-trivial, practical example of using our library for complex models.
>
> Nonetheless, we agree that results on the UCI dataset might be useful as well and so we have added them in section C in the new revision!

---

> > ### Comment · AnonReviewer3 · 2019-11-15
> > **Thank you for the further experiment**
> >
> > Thanks for providing the experiment with the UCI datasets, I find it to be a quite interesting demonstration of what your library can do, and as such I've bumped my score to weak reject.
> >
> > A couple of follow-up questions arise though that may be useful for further questions
> > 1) What depths of networks were typically found throughout the different datasets?
> > 2) Is there any suggestion as to why fixup initialization schemes ought to perform better?

---

> > > ### Author Response · Authors · 2019-11-15
> > > **Great questions!**
> > >
> > > >> A couple of follow-up questions arise though that may be useful for further questions
> > > >> 1) What depths of networks were typically found throughout the different datasets?
> > >
> > > Great question! We looked at it for FC Relu and Resnet + Fixup NTKs. While we do not think it really belongs in the paper, for your interest here is the plot: https://github.com/neural-tangents/neural-tangents/blob/master/iclr_figures/optimal_depth.pdf. We see that mostly single layer networks were selected for both architectures. It would be interesting to do future research, in Neural Tangents, to see if we can understand this!
> > >
> > > >> 2) Is there any suggestion as to why fixup initialization schemes ought to perform better?
> > >
> > > Residual networks cause gradients to explode which translates to poor conditioning of the NTK. It’s likely that fixup-style initialization schemes improve this conditioning. Again, this would be interesting research to perform using Neural Tangents.

---

> ### Author Response · Authors · 2019-11-15
> **[1/4] Addressing Major Comments**
>
> Thanks for your careful review of our work. We’re happy that you enjoyed the paper, found the library easy to use, and that you might use it in the future! We hope that this fact alone helps to convince you that researchers in the community might benefit from learning about Neural Tangents at ICLR.
>
> ----------------------------------------------------------------------------------------------------
> >>> While I really enjoyed reading the paper and believe that this library could be extremely practically useful, I vote to reject this paper because I do not feel that it has sufficient novelty to be a paper on its own in light of Lee et al, 2019. [...]
>
> >>> However, my primary concern with this paper is that it’s not sufficiently distinct from the previous work of Lee et al, 2019. After all, most of the experiments in that paper would have required the type of implementation that is described in greater detail in this paper.
>
> We would like to clarify the relationship between this work and Lee et al. 2019. We have added a summary of this discussion to section A in the revised version of the manuscript.
>
> TL;DR
>
> 1) Neural Tangents (NT) is emphatically different from the code of Lee et al, 2019 at the time of their paper submission (larger by thousands of lines of code (LOC)), and
>
> 2) The features in NT extend far beyond what was open-sourced with Lee et al, 2019 at the time of submission _and_ what could have been necessary for that paper.
>
>
> Specifically:
>
> 1) Raw code difference
> - Lee et al, 2019 at the time of submission: https://github.com/google/neural-tangents/tree/d42cc0f0281001d5885ed3969b61d69c8ccf4a15
> - Our codebase at HEAD: https://github.com/neural-tangents/neural-tangents
> - Most importantly, the +9,500/-2,500 LOC diff: https://github.com/neural-tangents/neural-tangents/compare/Lee_et_al_2019..master (we imported their code into a separate branch of our repo to show the difference)
>
> 2) Feature difference: at the time of the submission of Lee et al. 2019, their open-sourced code only had the following features (see github link above):
>
> - Linearization (equivalent of "nt.linearize"),
> - Single-sample estimate of the empirical ntk (equivalent of "nt.empirical_ntk_fn"),
> - Finite-time output mean evolution (equivalent of "nt.predict.gradient_descent_mse", "nt.predict.gradient_descent", "nt.predict.momentum").
>
> One can easily check that the vast majority of experiments in Lee et al. only used this functionality of Neural Tangents. Only a few experiments used very simple fully-connected ReLU kernels, which were not produced with Neural Tangents but an internal Tensorflow implementation (known via personal correspondence with the authors). These kernels are not unto themselves specific to Lee et al 2019; for example the arccosine kernel dates back to Cho and Saul in 2009.
>
> Any post-submission developments in their repository that were neither used nor necessary for their paper should not be considered published results but rather work concurrent to ours. [P.S. on an unrelated note, one could argue that even treating the original Lee et al, 2019 codebase as published is debatable, since they themselves, along with at least one other paper (https://arxiv.org/pdf/1905.13654v2.pdf) cited the code as a separate unpublished work]
>
> NT has all the features of Lee at al 2019 (with authors’ permission) at the time of submission, and:
> - Most notably, a high-level modular library "nt.stax" to specify and do inference with infinite NTK/NNGPs analytically for many NN layers. This is the highlight of the paper and was not used in / released with / necessary for Lee et. al, 2019.
> - Multi-device GPU/TPU support.
> - Parallelizable Monte-Carlo sampling of NTK and NNGP kernels.
> - Taylor series function expansion.
> - A richer suite of prediction functions including finite/infinite time NTK/NNGP mean/covariance prediction.
> - Unification of all of these features to work together seamlessly.
>
> We hope this, together with the updates in the text, helps clarify the contributions of our paper.

---

> > ### Comment · AnonReviewer3 · 2019-11-15
> > **Thank you for the clarification**
> >
> > Thank you for the clarification here - interesting that Hayou et al, 2019 cite the library as its own paper by late May (in my understanding).
> >
> > I agree that the current version of the codebase is considerably more developed than it was in early May but am still not particularly convinced that it's a distinct work given that it is fundamentally based in a similar language/functionality.
> >
> > Apologies for the terse reply.

---

> > > ### Author Response · Authors · 2019-11-15
> > > **Thank you for the quick response! A bit more on novelty**
> > >
> > > Dear Reviewer 3,
> > >
> > > Thank you for your very quick reply and for updating your score. We would still like to push back on the novelty aspect.
> > >
> > > >>> I agree that the current version of the codebase is considerably more developed than it was in early May but am still not particularly convinced that it's a distinct work given that it is fundamentally based in a similar language/functionality.
> > >
> > > While we certainly understand the initial confusion about the overlap with Lee et al, 2019, we believe that we have provided a substantial and precise summary of contributions that are specific to this submission, in both the rebuttal and updated text, backed up by a code diff (see https://openreview.net/forum?id=SklD9yrFPS&noteId=B1gt-wb3or ).
> > >
> > > We do not understand how using the same programming (or mathematical) language as Lee et al. 2019 can be an issue, since this is standard practice. We wish to reiterate that the key feature of the library (“nt.stax”, specification and computation of exact infinite-width kernels: https://github.com/neural-tangents/neural-tangents/blob/master/neural_tangents/stax.py ) is completely new in our library, was not released, was not used, and was not necessary for Lee et al. 2019. It was not “more developed”, but designed and implemented essentially from scratch.
> > >
> > > In this light, and again appreciating the time and thought you have given to our work, we would ask you to reconsider your score again. Thank you!

---

> ### Author Response · Authors · 2019-11-15
> **Reply**
>
> >>> Edit: post rebuttal, I'm bumping my score to a weak reject but would have minimal qualms if this paper were to
> >>> be accepted. I find the further experiments performed by the authors of very good quality overall, but I'm still not
> >>> particularly satisfied by their `argument that the codebase itself is distinct enough from separate related work. It's
> >>> unfortunately a bit hard, from a machine learning researcher side, to review the quality of a codebase in and of itself.
>
> We appreciate that it is difficult to review a codebase. To help with this we’d like to discuss the development of “neural_tangents.stax” which is the main focus of this work and was developed entirely after May. There has been a significant amount of work on NNGP and NTK methods over the past few years to compute NNGP and NT kernels for a growing set of architectures. In addition to noting that components ought to arbitrarily compose with one another (which I believe was not widely understood prior to this) development of "neural_tangents.stax" required arriving at efficient implementations for FC [1,2,3], CNN [4, 5, 6, 7], and pooling [5,7] kernels which were known in the literature along with FanOut, FanInSum, LayerNorm, parallel, and serial which were not explicitly known. As far as we are aware we are also the first to implement convolutions with arbitrary padding, shapes, and strides. Moreover, we wrote code to automatically parallelize this over large datasets. Finally, we note that our contribution also includes the neural tangent cookbook notebook which, as far as we are aware, includes the first computation of the mean and (now, thanks to reviewer 2) variance posterior prediction of the MSE loss. We believe that this represents a substantial research contribution.
>
> [1] Exponential expressivity in deep neural networks through transient chaos
> Poole et al. ; NeurIPS 2016
> [2] Deep Neural Networks as Gaussian Processes
> Lee et al. ; ICLR 2018
> [3] Neural Tangent Kernel: Convergence and Generalization in Neural Networks
> Jacot et al. ; NeurIPS 2018
> [4] Dynamical Isometry and a Mean Field Theory of CNNs
> Xiao et al. ; ICML 2018
> [5] Bayesian Deep Convolutional Networks with Many Channels are Gaussian Processes
> Novak et al.; ICLR 2019
> [6] Deep Convolutional Networks as shallow Gaussian Processes
> Garriga-Alonso et al.; ICLR 2019
> [7] On Exact Computation with an Infinitely Wide Neural Net
> Arora et al.; NeurIPS 2019

---

### Decision · Program_Chairs · 2019-12-19

**Decision:**

Accept (Spotlight)

**Comment:**

This paper presents a software library for dealing with neural networks either in the (usual) finite limit or in the infinite limit. The latter is obtained by using the Neural Tangent Kernel theory.

There is variance in the reviewers' scores, however there has also been quite a lot of discussion, which has been facilitated by the authors' elaborate rebuttal. The main points in favor and against are clear: on the positive side, the library is demonstrated well (especially after rebuttal) and is equipped with desirable properties such as usage of GPU/TPU, scalability etc. On the other hand, a lot of the key insights build heavily on prior work of Lee et al, 2019. However, judging novelty when it comes to a software paper is more tricky to do, especially given that not many such papers appear in ICLR and therefore calibration is difficult. This has been discussed among reviewers.

It would help if some further theoretical insights were included in this paper; these insights could come by working backwards from the implementation (i.e. what more can we learn about infinite width networks now that we can experiment easily with them?).

Overall, this paper should still be of interest to the ICLR community.